# The Inhibition of NLRP3 Inflammasome and IL-6 Production by *Hibiscus noldeae Baker f.* Derived Constituents Provides a Link to Its Anti-Inflammatory Therapeutic Potentials

**DOI:** 10.3390/molecules25204693

**Published:** 2020-10-14

**Authors:** Jean Claude Didelot Tomani, Vedaste Kagisha, Alembert Tiabou Tchinda, Olivia Jansen, Allison Ledoux, Luc Vanhamme, Michel Frederich, Raymond Muganga, Jacob Souopgui

**Affiliations:** 1School of Medicine and Pharmacy, College of Medicine and Health Sciences, University of Rwanda, P.O. Box 3286 Kigali, Rwanda; Jean.Claude.Tomani@ulb.ac.be (J.C.D.T.); kagishavedaste@gmail.com (V.K.); rmuganga@yahoo.fr (R.M.); 2Department of Molecular Biology, Institute for Molecular Biology and Medicine, Université Libre de Bruxelles, 6041 Gosselies, Belgium; Luc.Vanhamme@ulb.ac.be; 3Laboratory of Pharmacognosy, Centre for Interdisciplinary Research on Medicines (CIRM), University of Liège, B36, 4000 Liège, Belgium; ojansen@ulg.ac.be (O.J.); allison.ledoux@ulg.ac.be (A.L.); M.Frederich@uliege.be (M.F.); 4Laboratory of Phytochemistry, Centre for Research on Medicinal Plants and Traditional Medicine, Institute of Medical Research and Medicinal Plants Studies, Yaoundé P.O. Box 6163, Cameroon; talembert@gmail.com

**Keywords:** asthma, *Hibiscus noldeae*, bio-guided fractionation, Inflammasome, IL-1β, IL-6, THP-1, RAW264.7, macrophage

## Abstract

The activation of NOD-, LRR-, and pyrin domain-containing protein 3 (NLRP3) inflammasome and/or its components is associated with the physio-pathogenesis of many respiratory diseases including asthma, COPD (chronic obstructive pulmonary disease), SARS Cov-2 (severe acute respiratory syndrome coronavirus 2), and in several autoimmune diseases. *Hibiscus noldeae Baker f*. has been widely reported to be traditionally used in the treatment of different ailments, some of which are of inflammatory background such as asthma, wounds, headache, etc. However, the claims have not been supported by evidence at the molecular and functional levels. Here, we report on the bio-guided fractionation of *H. noldeae* and assessment of the inhibitory properties of some fractions and purified compounds on NLRP3 inflammasome and Interleukin 6 (IL-6). The activation of the NLRP3 inflammasome was determined by detecting the activity of caspase-1 and the production of Interleukin 1β (IL-1β) in Lipopolysaccharide (LPS) and ATP-stimulated Tamm-Horsfall Protein 1 (THP-1) macrophages, while the production of IL-6 was studied in LPS-stimulated RAW264.7 mouse macrophages. It was observed that hexane and ethyl acetate fractions of the crude extract of the aerial parts of *H. noldeae*, as well as caffeic acid, isoquercetin, and ER2.4 and ER2.7 fractions revealed significant inhibitory effects on Caspase-1 activities, and on IL-1β and IL-6 production. The ER2.4 and ER2.7 fractions downregulated the production of IL-1β and IL-6, in a similar range as the caspase-1 inhibitor AC-YVAD-CHO and the drug Dexamethasone, both used as controls, respectively. Overall, our work does provide the very first scientific based evidence for *Hibiscus noldeae* anti-inflammatory effects and widespread use by traditional healers in Rwanda for a variety of ailments.

## 1. Introduction

Respiratory diseases, with asthma being the most important, are classified among the major categories of non-communicable diseases and their prevalence accounted for more than 1 billion patients worldwide in 2018 [1,2]. Asthma is a chronic inflammatory disease resulting from the malfunction of the interaction between the airways and the immune cells [3,4]. Inflammation is considered to be a physiological defense mechanism which helps the body to protect itself from microbial infections, burn, toxic chemicals, allergens, or other harmful stimuli. Depending on the number or the degree of events and mediators linked to that reaction, inflammatory reaction can trigger, maintain, or aggravate many diseases [5]. Sometimes the delicate balance between immune-mediated clearance and noxious stimuli is troubled and this correlates with inadvertent self-harm from chronic inflammation. As a result, the immune system inappropriately leads to inflammation against its own cells, resulting in different chronic inflammatory diseases including respiratory diseases such as asthma and chronic obstructive pulmonary disease (COPD) [6]. Recently, the complicated form of SARS Cov-2 (severe acute respiratory syndrome coronavirus 2) viral pneumonia outbreak was reported to be associated with a plethora of cytokines wreaking havoc throughout the body often vividly referred to as “cytokine storm”, which triggers a violent inflammatory immune response that contributes to respiratory problems, multiple organ failure, and finally death in severe cases of SARS-CoV-2 infection [7,8,9,10]. Several studies have reported the role of different inflammatory cells and mediators in the pathogenesis of asthma where the volume and nature of involved inflammatory mediators determine the severity and exacerbations of the disease. Depending on those inflammatory patterns, there are two main subtypes of asthma, the eosinophilic (diagnosed by the expression of Th2-derived cytokines) and the neutrophilic (diagnosed by the expression of Th17-derived cytokines and chemokines) [11]. The latter asthma subset is characterized by a high proportion of neutrophils recruited by inflammatory cytokines like Interleukin 1β (IL-1β), Interleukin 6 (IL-6), and Tumor Necrosis Factor Alpha (TNF-α). It is well known that inflammasomes, multiprotein cytoplasmic complexes activating inflammatory responses, are involved in the upregulation of these cytokines, particularly the IL-1β through the activation of caspase-1 [12,13]. Among these inflammasomes, NOD-, LRR-, and pyrin domain-containing protein 3 (NLRP3) is the most widely studied and its properties and functions are the best characterized [14]. In fact, in response to a microbial infection or injury, the NLRP3 inflammasome is activated and involved in the host protection by inducing immune responses to clear injury or microbial invasion [13,14]. The activation of NLRP3 inflammasome requires two distinct signals. The first is a pro-inflammatory priming step, mediated by toll-like receptors (TLR), that induces the upregulation of the NF-κB transcription factor and promotes the expression of pro-IL-1β and the NLRP3 inflammasome component. The second signal is an activation step that induces the assembly of the NLRP3/ Apoptosis-associated speck-like protein containing a caspase recruitment domain (ASC)/pro-caspase-1 complex, ending up in the proximity auto-cleavage and maturation of caspase-1 which, in turn, induces a caspase-1-mediated maturation of pro-IL-1β and release of active IL-1β [15]. Several pieces of evidence from research suggest that inflammasomes and/or their components and effectors are associated with the physio-pathogenesis of asthma and other respiratory diseases including COPD, SARS CoV-2, etc. [15,16,17,18]. In fact, it has been reported that the excessive release of active IL-1β and ASC (apoptosis-associated speck-like protein containing a caspase recruitment domain) specks leads to the induction of neutrophil-dominated inflammatory responses in the airways that contributes to the pathogenesis of COPD and neutrophilic asthma [16,17].

The neutrophilic asthma is the most severe and has been reported to be refractory to the corticoid-based treatment [16]. Despite this resistance, the use of corticosteroids remains the basic treatment for several respiratory diseases involving cytokine storms. A recent clinical trial report revealed that the treatment with Dexamethasone significantly alleviated COVID-19 disease severity of hospitalized patients who were under invasive mechanical ventilation or oxygen [19]. Regarding the neutrophilic asthma, drugs that alleviate disease severity include specific monoclonal antibodies against key mediators of the inflammatory pathway [20] but the associated treatment cost is not affordable for low and middle income countries (LMICs). Hence, urgent efforts are needed to identify new, effective, affordable, and safe anti-inflammatory drugs. A reliable therapeutic source which remains poorly explored is the traditional pharmacopeia. Previously, our investigations on the anti-asthmatic activities of herbal recipes from Rwandese traditional healers revealed a candidate recipe with significant inhibitory properties on the activation of caspase-1 using Caspase-Glo 1 Inflammasome assays [21]. *Hibiscus noldeae*, from the *Hibiscus* genus in the *Malvaceae* family, was one of the plants composing the recipe. Plants from the same *Hibiscus* genus, including *H. acetosella*, *H. sabdariffa*, and *H. vitifolius* have been reported to have anti-inflammatory potentials and some of the responsible molecules, such as gossypin, caffeoylhydroxycitric acid, delphinidin-3-sambubioside, and neochlorogenic acid have been isolated [22,23,24]. In contrast, no such studies has been reported on *H. noldeae*, although used by traditional healers in the treatment of asthma, urogenital infections, postpartum abdominal pain, swelling of the legs, cleaning of babies, facilitation of delivery, sprains, headache, wounds, women infertility, abortion prevention, etc. [21].

In this study, we report on the bio-guided fractionation of *H. noldeae* and assessment of the inhibitory properties of some fractions and purified compounds on NLRP3 inflammasome and IL-6. The activation of the NLRP3 inflammasome was determined by detecting the activity of caspase-1 and the production of IL-1β in Lipopolysaccharide (LPS)- and ATP-stimulated THP-1 macrophages, while the production of IL-6 was studied in LPS-stimulated RAW264.7 mouse macrophages, a well-known standard method. It was observed that hexane and ethyl acetate fractions of the aerial parts of *H. noldeae*, as well as caffeic acid, isoquercetin, and ER2.4 and ER2.7 fractions revealed the highest inhibitory effects on Caspase-1 activities, and on IL-1β and IL-6 production. The ER2.4 and ER2.7, our fractions of most interest, downregulated the production of IL-1β and IL-6, in a similar range as the caspase-1 inhibitor AC-YVAD-CHO and the drug Dexamethasone all used as controls, respectively. Overall, our findings pave the way for the first time on the scientific based evidence on *Hibiscus noldeae* activities underlying its potential role in the treatment of inflammatory diseases and asthma in Rwanda.

## 2. Results

*Hibiscus noldeae* (*H. noldeae*) has been studied by virtue of its attribute as one of the medicinal plants used by the traditional healers in Rwanda to manufacture their remedies. In the context of asthma treatment, the recipe, containing 7 different plants, that revealed the best anti-inflammatory activity [21] comprised *H. noldeae* in major proportion (35%). We therefore hypothesized that this medicinal plant may harbor the therapeutic constituents.

### 2.1. Hexane and Ethyl Acetate Fractions Revealed Similar HPLC Profiles for the Constituents of the Crude Extract of the Aerial Parts of H. noldeae

A bio-guided fractionation approach was employed and as shown in Figure 1, a complex chemical profile for hexane and ethylacetate fractions was observed in the High performance Liquid Chromatography with Diode-Array Detection (HPLC-DAD) analysis, indicating the presence of different major substances of different polarities. Many peaks eluted between 20 and 40 min, a fact that demonstrates the presence of medium polarity molecules, most of which are major compounds. The transposition of the analytical method to the preparative HPLC allowed for the isolation of five of the seven major compounds from ethylacetate and hexane fractions of *H. noldeae*. Three of the five isolated compounds were pure and were identified by chromatographic and/or spectroscopic comparison with data from literature. This led to the elucidation of one flavone glycoside, namely quercetin 3-*O*-glucopyranoside, along with two polyphenols, caffeic acid and caffeoyl-hydrocitric acid (Appendix A).

### 2.2. Hexane and Ethyl Acetate Extracted Constituents of Hibiscus noldeae Inhibit the Activation of the NLRP3 Inflammasome in THP-1 Macrophages

Prior to the investigation of anti-inflammatory properties of the compounds of interest, their effect on the cell viability on both THP-1 monocytes and derived macrophages was first determined. Cells were treated with different concentrations (0, 25, 50, and 100 µg/mL) for 24 h, and the cell viability was determined using CellTiter-Glo Luminescent Cell Viability Assay kit (Promega Co., Madison, WI, USA). Up to 25 µg/mL, all tested compounds did not exhibit cytotoxicity to both THP-1 and RAW264.7 cell lines (Appendix A). Our anti-inflammatory activities were therefore determined within this range.

To investigate whether the isolated compounds and fractions of interest possessed potential NLRP3 inflammasome inhibitory effects, their influence on the NLRP3 inflammasome activation, in LPS and ATP-stimulated THP-1-derived macrophages was assessed. The levels of caspase-1 activities in the culture media were determined using the Caspase-Glo^®^ 1 Inflammasome Assay (Promega Co., Madison, WI, USA). As shown in Figure 2 below, it was observed as expected that LPS and ATP treatment significantly (*** *p* < 0.0001) increased caspase-1 activities in the cell culture supernatant when compared with the untreated cells. However, when cells were pretreated with the isolated compounds and fractions of interest in graded concentrations ranging from 5 to 20 µg/mL, the caspase-1 activities were reduced by at least 34%. The tested compounds revealed their maximal inhibitory activities at 5 µg/mL, except for the ER2.4 fraction that showed a dose-dependent inhibition of the caspase-1 activation (Figure 2).

Next, the production of IL-1β was investigated using the human IL-1β (enzyme linked immunosorbent assay) ELISA kit (Invitrogen/Thermofisher, Waltham, MA, USA). As controls, two distinct reference inhibitors, namely Ac-YVAD-CHO (YVAD) which specifically acts on caspase 1 and Dexamethasone (Dex) which downregulates the NF-κB pathway were used. As results, it was observed as expected that LPS and ATP treatment significantly (*** *p* < 0.0001) increased the production of IL-1β in the cell culture supernatant when compared with the untreated cells (Figure 3A). When cells were first treated with *H. noldeae* constituents or the controls, the production IL-1β was significantly inhibited. Reductions of more than 70% could be revealed in some cases (Figure 3A). Moreover, it was observed that, at equivalent dose of 20 µg/mL, ER2.7 fraction and Dexamethasone showed similar inhibitory effects, while at the same concentration other tested constituents of *H. noldeae* revealed inhibitory activities comparable to the caspase 1 reference inhibitor YVAD. Isoquercetin was observed to have the lowest inhibitory effect on IL-1β production (Figure 3A).

It is well-known that the activation of the NLRP3 inflammasome induces the activation of caspase-1 that promotes the production of Gasdermin D. The N-terminal domain of Gasdermin D oligomerizes to form cytolytic pores and leads to the programmed cell death called “pyroptosis” [25]. Therefore, we investigated the effect of our compounds and fractions of interest on inflammasome activation-induced cell death by measuring the release of lactate dehydrogenase (LDH) in the culture supernatants. As expected, the pretreatment of cells by our test compounds significantly inhibited ATP-induced pyroptosis (Appendix A).

NLRP3 inflammasome-associated IL-1β production is a two-step process: a priming signal that promotes Pro-IL-1β and NLRP3 synthesis and a secondary signal that induces the inflammasome assembly and the subsequent caspase-1 activation [26]. We investigated the effect of the crude extract and fractions of interest on the expression of Pro-IL-1β and Pro-Caspase-1 by Western blot (WB). We observed as shown in Figure 3B that cells pretreatment with ER2.4 and ER2.7 fractions led to the reduced Pro-Caspase-1 amount marked by the faint WB-signals. This suggest that our fractions of interest inhibited the expression of this protein by impacting the first priming step of the NLRP3 inflammasome activation or induced its degradation by a mechanism to be clarified. Moreover, the Pro-IL-1β production appeared induced as marked by the width of the WB-signals (Figure 3B). Knowing the critical role of Caspase-1 in the inflammasome activation, these observations suggest that the reduced expression of the Pro-Caspase by both ER2.4 and ER2.7 fractions may in fact be acting to inhibit the upregulation of Caspase-1, hence block the Pro-IL-1β cleavage. In addition, treatment with Dexamethasone, the crude extract and the EtoAc fraction revealed reduced Pro-IL-1β WB-signals. While the effect of Dexamethasone is associated with the inhibition of the NF-κB activities leading to the repression of Pro-IL-1B production, the effects of the crude extract and EtoAc fraction could results from the dilution of the ER2.4 and ER2.7 fractions or likely their effects on NF-kB activities by their constituents to be identified.

### 2.3. Hexane and Ethyl Acetate Fractions and Purified Compounds from H. noldeae Inhibit IL-6 Production

Interleukin 6 (IL-6) like IL-1β is a biomarker of inflammasome activation. Hence, the effect of the fractions and purified compounds from *H. noldeae* on the production of IL-6 in RAW264.7 macrophages supernatants by ELISA was investigated. Dexamethasone at concentration of 20 µg/mL was used as control. Treatment of RAW264.7 cells with LPS alone resulted in a significant increase in this cytokine production as compared to the control group. As expected, a pre-treatment of cells with Dexamethasone reduced about 72% of the production of IL-6. Also, it was observed that the levels of IL-6 production were significantly (*** *p* < 0.0001) decreased upon pre-treatment of cells with *H. noldeae* constitutuents. Moreover, while a saturation of inhibitory effects was already observed at either 5 or 10 µg/mL with the ER2.4 fraction or with the crude extract and the EtoAc fraction, a dose-dependent inhibition of IL-6 production was revealed by the caffeic acid and ER2.7 fraction pre-treatment. At 20 µg/mL, the inhibitory effect of ER2.7 fraction was similar to that of Dexamethasone (Figure 4).

## 3. Discussion

An important body of studies have proven the associations between neutrophilic asthma and increased IL-1β responses induced by the excessive NLRP3 inflammasome activation. This raises the possibility of targeting different inflammasome components as substantial therapeutic targets in neutrophil-dominated forms of severe, steroid-refractory asthma [16]. The analysis of inflammasome formation is mostly investigated through the monitoring of downstream processes and outcomes, the presence of cleaved caspase-1 enzyme being the most widely used method, as well as the caspase-1 activity on substrates such as pro-IL-1β, pro-IL-18, and pro-gasdermin-D [14]. In view of determining the inhibitory properties of the *H. noldeae* constituents of interest on NLRP3 inflammasome, their effect on either caspase-1 activation using the Caspase Glo 1 assays (Promega, Madison, WI, USA) and/or IL-1β production as hallmarks of the NLRP3 inflammasome was evaluated. In fact, it is well known that the inflammasome activation induces a cascade of events including the recruitment of pro-caspase-1 and its cleavage into a corresponding active caspase-1 enzyme that also cleaves pro-IL-1β into its mature form, IL-1β [14].

In our previous study, among the plant recipes evaluated, R19Sz was the most potent in inhibiting caspase-1 activity [21]. *Hibiscus noldeae*, one of the plant’s components of the recipe, has been cited by three different traditional healers during our investigation, indicating its possible importance in the treatment of asthma. In addition, few studies reported on the traditional use of the plant in the treatment of different ailments, some of which are of inflammatory background such as asthma, wounds, headache, etc. [21,27,28,29]. An important question was then to determine whether *H. noldeae*, would have inhibitory properties on inflammasome or on IL-6 production, a well-known downstream target of IL-1β. A bioguided fractionation approach to address this question was employed. Many of the isolated compounds (Figure 1 and Appendix A) were polyphenols, secondary plant metabolites proven to exert various biological properties including anti-bacterial, anti-inflammatory and anti-allergic activities [30,31,32]. As far as inflammation is concerned, cell death could also lead to the activation of the inflammatory pathway in a manner similar to the inflammasome activation [33,34]. Therefore, plant crude extracts, fractions and purified compouds were first tested for eventual cytotoxicity allowing to select the range of working sample concentrations in such a way that their pharmacological effects could not be biased.

Within a plant extract, there can be hundreds of molecules, some of which are active individually or act synergistically against a given disease. In most of the cases, major compounds are responsible of the activity of the extract though it is not always the case [35,36]. In this study, an important question was to determine whether the inhibition of NLRP3 inflammasome resulted from a single compound or from a combined effect of different compounds. We opted first to investigate the effect of cell pre-treatment with our various samples on the caspase-1 activities and observed an average reduction of about one-third which formed a plateau at the tested concentration ranges as shown in (Figure 2). Such reduction suggested the presence of an inhibitor of caspase-1 in the plant extract. Indeed, caffeic acid, purified from *H. noldeae* in this work, was reported to interact with and inhibits Caspase-1 and ASC activation [36]. The plateau could be explained by the fact that a saturated quantity of caffeic acid is already in the lowest sample concentration used (5 µg/mL).

The findings that our plant products of interest had inhibitory effects on the caspase-1 activities prompted us to assess the production of IL-1β in the supernatant from the different samples. Consistent with the inhibition of the caspase-1 activities all the tested crude extract, fractions and purified constituents of *H. noldeae* revealed significant inhibition of IL-1β production (Figure 3A). To ensure that *H. noldeae* constituents acted on the inflammasome activation pathway, we further investigated their effects on pyroptosis. In fact, the activation of the NLRP3 inflammasome induces the activation of caspase-1 that promotes the production of Gasdermin D and the latter is associated to the programmed cell death called “pyroptosis” [25]. In contrast, compared to the production of IL-1β, the effect of Dexamethasone was less than that of ER2.4 and ER2.7 fractions (Appendix A). This may be explained by the fact that Dexamethasone is able to act at the transcriptional level or to selectively induce the degradation of IL-1β transcripts after their production [37]. Thus, the additional effect of these both mechanisms justifies its enhanced inhibition of IL-1β production in the supernatant. However, at the highest concentration tested (20 µg/mL) an unexplained ease of the inhibitory effect was observed with both fractions. Such unexpected reactions deserve further investigation during the characterization of ER2.4 and ER2.7 fractions. 

NLRP3 inflammasome-associated IL-1β production is a two-step process. Our findings on both pyroptosis assays (Appendix A) and IL-1β production (Figure 3A) suggest that *H. noldeae* derived products interfere with the NLRP3 inflammasome signaling, to block cell death (pyroptosis) and IL-1β production. However, the Western blot results showed that ER2.4 and ER2.7 fractions repressed the expression of Pro-Caspase-1 (Figure 3B). Capsase-1 is a critical component of the inflammasome and the reduced expression of this enzyme by fractions from *H. noldeae* indicates that ER2.4 and ER2.7 may in fact be acting to inhibit signal 1 (priming step) of inflammasome activation by specifically blocking upregulation of Caspase-1. In our future research, the mode of action of the molecules contained in these fractions will be elucidated.

Several studies have reported in vivo anti-inflammatory properties of caffeic acid and isoquercetin as well as several corresponding derivatives [36,38,39]. In this study, as suggested by our results, several polyphenolic compounds detected in the plant extracts of *H. noldeae* likely contribute to the inhibitory property of the plant on the inflammasome expression. However, the activity of the plant remains inferior to the activity of purified caffeic acid as well as the concentrated fractions ER2.4 and ER 2.7. This suggests that the fractionation of the plant extracts led to the enhancement of their inflammasome inhibition potency. This is in accordance with previous studies that reported the beneficial effect of fractionation in concentrating polyphenolic compounds, leading to enhanced biological activities [40,41]. We thus argue that the bioactive compounds in fraction ER2.4, ER2.7 together with the well-known caffeic acid and isoquercetin are the main compounds involved in the regulation of the NLRP3 inflammasome.

In the present study, the effect of our plant samples on the production of IL-6 in RAW264.7, a murine cell line widely used for the in vitro production of IL-6, was also investigated. This cytokine is a known downstream target of IL-1β and is consistently increased in serum from patients with NLRP3 inflammasome-mediated conditions. In fact, IL-1β has been reported to be one of the most potent activators of IL-6 production, both in vitro and in vivo, suggesting that IL-6 could be a therapeutic target in the treatment of IL-1β-mediated inflammation [42]. In addition, abnormal expression of IL-6 has been associated with the pathogenesis of a variety of human diseases, including asthma, cancers and autoimmune diseases such as rheumatoid arthrosis, and recently with COVID-19 [43,44]. In asthma, it has been reported that IL-6 is linked with mixed eosinophilic–neutrophilic bronchitis during exacerbations of the disease and worse pulmonary function in humans [44]. Moreover, cytokine-targeting therapies are gaining an increasing popularity in the treatment of inflammatory diseases including asthma and other respiratory diseases. With the Covid-19 outbreak pandemic, it has been reported that dexamethasone and monoclonal antibody drugs targeting IL-6 (tocilizumab) reduced the mortality rate [19,45,46]. In the light of the above-mentioned role of IL-6 in different pathologies, we investigated the effect of the *H. noldeae* constituents of interest on its production in RAW264.7 cell line. As shown in Figure 4, results obtained indicate that fractions and partially purified ER2.4 and ER2.7 compounds from the plant exhibited similar trends as on IL-1β production (Figure 3A). These findings allowed to argue that this medicinal plant is an important drug target for ‘cytokine storm’- related diseases such as the neutrophilic asthma and the COVID-19 pandemic.

## 4. Materials and Methods

### 4.1. Collection of Plant Materials

Aerial parts of *Hibiscus noldeae were* collected from the arboretum of Ruhande, in Huye district, Southern province in Rwanda. The plant species was identified with reference to herbarium specimens and various literature resources by Nshutiyayesu Samuel (from the University of Rwanda, Huye, Rwanda) and the authentication and ID verification by Dr. Olivier Lachenaud (from the Botanic Garden Meise/Meise, Belgium) as previously reported [21].

### 4.2. Isolation and Structure Elucidation of Major Compounds

#### 4.2.1. Crude Extract Preparation and Fractionation

The aerial parts of *H. noldeae* were cleaned of dust and debris and washed gently with tap water, and air-dried under shade. The plant material was then pulverized with a grinder. A total amount of 500 g of the powder was extracted with the mixture of methanol: dichloromethane (3:1) as previously described [21]. The dried crude extract (55 g) was triturated in distilled water and then subjected to a liquid-liquid partitioning process using n-hexane and ethyl acetate successively. In each case, the mixture to be partitioned was vigorously shaken for 10 min, rested for 20 min, and then separated in a separatory funnel. Each partition was conducted three times and the same fractions were combined and dried-up using rotary evaporator. At the end, the non-soluble part was dissolved in methanol to make the methanol fraction.

#### 4.2.2. High-Performance Liquid Chromatography (HPLC) Analysis and Purification

Major compounds from two fractions showing the highest inhibitory activity on NLRP3 inflammasome components, namely hexane and ethylacetate fractions, were further isolated on preparative HPLC. To this end, an in-house HPLC method used to screen the major components of crude extracts was used. In this regard, analytical separation was carried out on a Hypersil ODS^®^ RP18 column (250 × 4.6 µm; particle size 5 µm; Thermofisher Scientific, Waltham, MA, USA). All samples were dissolved in methanol HPLC-grade, filtered through a 0.45 μm pore size filter membrane and analyzed on an Agilent (Santa Clara, CA, USA) 1100 HPLC machine. Samples were eluted with a nonlinear gradient method with acetonitrile (solvent A) and 0.05% trifluoroacetic acid in ultra-pure water (solvent B) (Table 1). The column temperature was maintained at 25 °C. Then, 20 μL of each sample was injected into the HPLC-UV/DAD system and the analysis, performed at a flow rate of 1.0 mL/min, was monitored at 210, 254, 288, and 350 nm. 

Preparative HPLC analysis was carried out on a Variant PrepStar machine. All extracts (500 mg) were dissolved in methanol and then diluted with water. The amount of methanol was not allowed to exceed 30%. Samples were filtered through a 0.45 µm filter membrane before injection. The mobile phase consisted of trifluoroacetic acid (TFA) 0.05% in ultrapure water (A) and acetonitrile (B). The Table 2 gives the gradient used to separate the major compounds identified with HPLC analytical method described above. This gradient was obtained by transposing the analytical HPLC methods to preparative HPLC using HPLC calculator. The flow rate was 15 mL/min and the separation was monitored at 254 and 350 nm. Structural identification of isolated compounds was performed by 1D and 2D Nuclear Magnetic Resonance (NMR) and infrared spectroscopy (IR). The detection was followed at 254 and 350 nm.

### 4.3. Cell Culture and Viability Assay

Human leukemia THP-1 monocyte and RAW264.7 murine cell lines were kindly received from the Institute for Medical Immunology/BioPark/ULB and were cultured in RPMI 1640 (Life Technologies, Waltham, MA, USA) and DMEM, respectively. All media were supplemented with 10% heat-inactivated Fetal Bovine Serum (FBS; Life technologies), 1% glutamine, and 1% of penicillin-streptomycin (Life technologies) and 1% Sodium pyruvate (for RAW264.7 only) and maintained at 37 °C under 5% CO_2_ in a humidified atmosphere. Cells were passaged at least 5 times before any experiment. For the determination of cell densities, the cell counting was done using a Neubauer counting chamber with trypan blue (0.4% in PBS; pH = 7.4; Life technologies) staining. Before investigating the anti-inflammatory activity of different fractions, their effect on the cell viability (THP-1, monocytes and derived-macrophages, and RAW264.7) was screened as previously reported [21].

### 4.4. Caspase-1 Inhibition Assays

The evaluation of caspase-1 activity was done using the Caspase Glo 1 inflammasome assay (Promega Co. Madison, WI, USA). Briefly, THP-1 cells were grown for 2 days in 24-well plates to a density of 2 × 10^5^ cells/well in the presence of 10 nM PMA at 37 °C in a humidified 5% CO_2_ incubator. PMA-supplemented medium was then discarded and macrophages were pretreated with fresh medium containing fractions at different concentrations for 1 h. Dexamethasone at a concentration of 20 µg/mL or 20 µM AC-YVAD-CHO treatments were used as controls. Cells were further incubated with LPS (Sigma, Saint-Louis, MO, USA) (at a final concentration of 1 µg/mL) for 3 h and then with 5 µM ATP (Invitrogen/Thermofisher, Waltham, MA, USA) for additional 1 h. The caspase-1 activity was measured using the Caspase-Glo 1 Inflammasome Assay (Promega Co., Madison, WI, USA) according to the manufacturer’s instructions. The 24-well plate containing treated cells was removed from the incubator and 50 µL of the supernatant in each well were transferred to the corresponding well of a new white 96-well plate. An aliquot of 50 µL of the Caspase-Glo 1 reagent was then added to each well and gently mixed on a plate shaker at 300 rpm for 30 s. The mixture was then incubated for 1 h at room temperature before measuring the luminescence on a Berthold Technologies luminometer.

### 4.5. THP-1-Derived Macrophage Treatment and Measurement of IL-1β

The culture supernatant of stimulated the THP-1 macrophage was used for the measurement of the cleaved form of IL-1β. The concentration of IL-1β in the culture supernatant was measured using a human IL-1β ELISA kit (Invivogen/Thermofisher, Waltham, MA, USA) according to the manufacturer’s instructions.

### 4.6. RAW264.7 Cell Treatment and Measurement of IL-6 Cytokine Production

In all experiments, RAW264.7 cells were sub-cultured in a 24-well plate and incubated for 24 h at 37 °C and 5% CO_2_ to acclimate before treatment. They were then pretreated with test compounds, at indicated concentrations, for 3 h prior to 18 h stimulation with 1 µg/mL LPS. Cell-free supernatants were harvested and stored at −70 °C until analysis. The concentration of IL-6 in the culture supernatant was measured using an ELISA kit (BD Biosciences, Franklin Lakes, NJ, USA) according to the manufacturer’s instructions. 

### 4.7. Pyroptosis Inhibition Assay

The evaluation of pyroptosis inhibition was done using the LDH-Glo^TM^ Cytotoxicity Assay (Promega Co. Madison, WI, USA). Briefly, THP-1 cells were cultured for 2 days in 96-well plates to a density of 5 × 10^4^ cells/well in the presence of 10 nM PMA at 37 °C in a humidified 5% CO_2_ incubator. PMA-supplemented medium was then discarded and macrophages were pretreated with fresh medium containing fractions at different concentrations for 1 h. Dexamethasone at a concentration of 20 µg/mL or 20 µM AC-YVAD-CHO treatments were used as controls. Cells were further incubated with LPS (Sigma, Saint-Louis, MO, USA) (at a final concentration of 1 µg/mL) for 3 h and then with 5 µM ATP (Invitrogen/Thermofisher, Waltham, MA, USA) for additional 1 h. Then, 2 μL of 10% Triton X-100 was added to 100 μL of Vehicle-treated Cells Control for 5 min before collecting the samples for LDH detection. Supernatants were harvested and 20-fold diluted in the LDH storage buffer and kept at –70 °C until analysis. The LDH activity was measured using the LDH-Glo ^TM^ Cytotoxicity Assay (Promega Co., Madison, WI, USA) according to the manufacturer’s instructions. The % Cytotoxicity = 100 × (Experimental LDH Release − Medium Background)/(Triton X-100 LDH Release Control − Medium Background).

### 4.8. Immunoblotting Analysis

For Western blot analysis, THP-1 cells (5 × 10^5^ cells/well) were differentiated in a 12-well plate by 10 nM PMA for 2 days and then treated as stated above. Cells were washed with 600 µL of an ice-cold PBS, and directly lysed with a reporter lysis buffer (Promega) and then centrifuged at 11,000 rpm (round per minute) for 15 min at 4 °C. The supernatants were kept at −70 °C until analysis. The protein quantitation in the supernatant was quantified by the Bradford assay. An equivalent amount (15 µg of protein) of each sample was separated by sodium dodecyl sulfate polyacrylamide gel electrophoresis (SDS-PAGE) and electro-transferred onto Nitrocellulose membranes. The membranes were immunoblotted by anti-ProIL-1b (Cell signaling Technology, Danvers, MA, USA) or anti-Pro caspase-1 (abcam) according to the manufacturer’s instructions. Membranes were then stripped, reprobed with anti-β-Actin antibodies (Cell signaling Technology) and exposed again to detect the internal β-Actin standard. Immunoblots were visualized using a chemiluminescence Odyssey Imaging System (Odyssey^®^ Fc, Bad Homburg, Germany). To scan and quantify the immunoblots, we used the Image Studio Lite Ver 5.2 software (Li-Cor Biotechnology-GmbH, Bad Homburg, Germany).

### 4.9. Data Presentation and Statistical Analysis

Each test was done in triplicates and all the experiments were reproduced at least two times independently. For each ELISA experiment, the amount of IL-1β or IL-6 produced was considered as 100% and the remaining treatments was calculated as (Treatment/LPS) × 100. The results are presented as the mean ± SD from two or three independent experiments (*n* = 3), each done in triplicate. Data were analyzed using GraphPad Prism 5 and Statistical Analysis Data expressed as the mean ± standard deviation. Comparisons of data between groups were performed by one-way ANOVA followed by Benforroni’s multiple comparison test. The *p*-values less than 0.05 were considered to be statistically significant.

## 5. Conclusions

Considering the importance of inflammasome components in the severity of respiratory diseases in general and in asthma in particular, our preliminary results at the molecular and functional levels suggest that *H. noldeae* and/or its constituents may play a role in the management of asthma and other respiratory diseases if the ex vivo results obtained can be reproduced in vivo. Additional purification and characterization of ER2.4 and ER2.7 fractions are of prime interest for our future study.

## Figures and Tables

**Figure 1 molecules-25-04693-f001:**
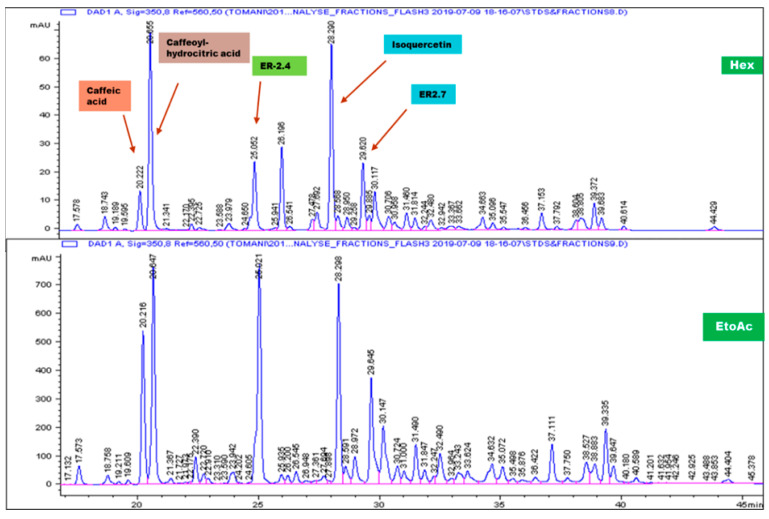
High performance Liquid Chromatography (HPLC) Chromatograms of the hexane (Hex) and ethylacetate (EtoAc) fractions of *H. noldeae*. An in-house HPLC method used to screen the major components of crude extract was used. Analytical separation was carried out on a Hypersil ODS^®^ RP18 column (250 × 4.6 µm; particle size 5 µm; Thermofisher Scientific, Waltham, MA, USA). All samples were dissolved in methanol HPLC-grade, filtered through a 0.45 μm pore size filter membrane and analyzed on an Agilent 1100 HPLC machine. Samples were eluted with a nonlinear gradient method with acetonitrile and 0.05% trifluoroacetic acid in ultra-pure water. The column temperature was maintained at 25 °C. Then 20 μL of each sample were injected into the HPLC-UV/DAD (Diode-Array Detection) system and the analysis, performed at a flow rate of 1.0 mL/min, was monitored at 210, 254, 288, and 350 nm.

**Figure 2 molecules-25-04693-f002:**
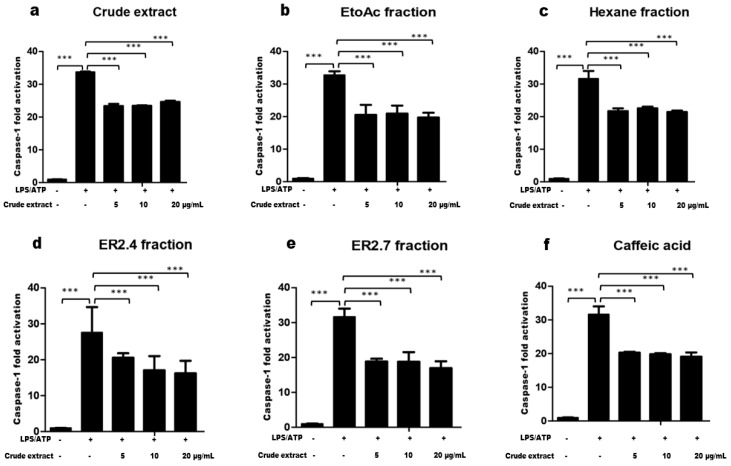
Crude extract (**a**) and Hexane (**c**) and ethylacetate (**b**) fractions as well as the caffeic acid (**f**) and semi-purified fractions, ER 2.4 (**d**) and ER 2.7 (**e**), thereof exhibited significant inhibition of caspase-1 in THP-1 derived macrophages. Phorbol Myristate Acetate (PMA)-differentiated cells were pretreated by corresponding compounds for 1 h, and then primed by Lipopolysaccharide (LPS) (1 µg/mL) for 3 h. Cells were then stimulated by ATP (5 mM) for additional 1 h. The supernatants were harvested and kept at −70 °C before analysis. Then, 50 µL of the culture supernatant were used to determine the caspase-1 activity using Caspase-Glo 1 inflammasome assay (Promega Co., Madison, WI, USA). The results are presented as the mean ± SD from independent experiments (*n* = 2), each done in triplicate. A one-way ANOVA, followed by Benforroni’s multiple comparison test, was used to compare the sample groups. *** means Caspase-1 fold activation changes with a *p* < 0.0001.

**Figure 3 molecules-25-04693-f003:**
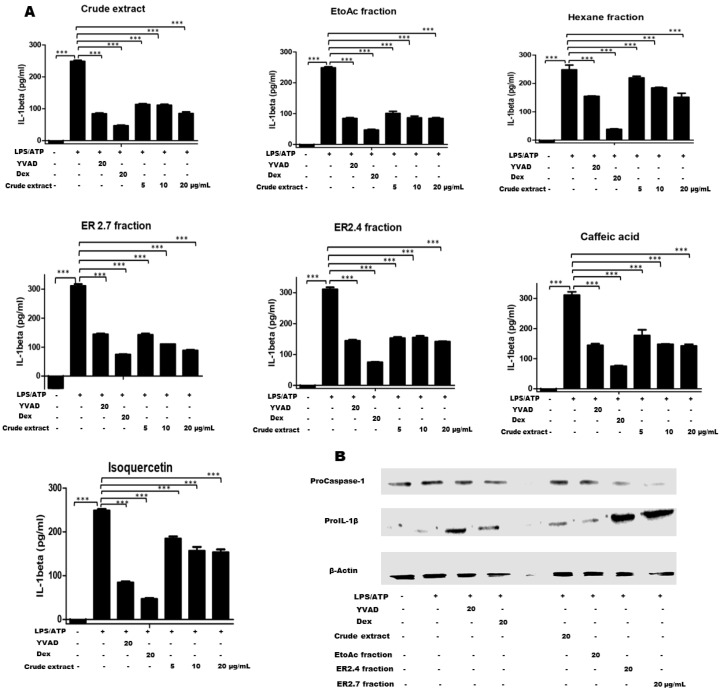
Inhibition of IL-1β production by *H. noldeae* constituents. (**A**) PMA-differentiated cells were pretreated with the indicated plant constituents at the concentration range shown for 1 h, and then primed by LPS (1 µg/mL) for 3 h. Concentrations of 20 µM YVAD and 20 µg/mL Dex were used as controls. Cells were then stimulated by ATP (5 mM) for additional 1 h. Supernatants were harvested and kept at −70 °C before analysis. For the ELISA analysis, samples were diluted 10 times in the assay diluent. The concentration of IL-1β in the culture supernatant was measured using a human IL-1β Enzyme linked immunosorbent assay (ELISA) kit (Invitrogen/Thermofisher, Waltham, MA, USA) according to the manufacturer’s instructions. For each experiment, the amount of IL-1β produced without any pretreatment was considered as 100% and the remaining treatments were calculated as (Treatment/LPS) × 100. The results are presented as the mean ± SD of a triplicate from a representative of three independent experiments (*n* = 3). A one-way ANOVA, followed by Benforroni’s multiple comparison test, was used to compare the sample groups. *** stands for *p* < 0.0001. (**B**) Immunoblotting of cell lysates by anti-ProIL-1β (Cell signaling Technology, Danvers, MA, USA) or anti-Pro caspase-1 (Abcam, Cambridge, UK) was done twice according to the manufacturer’s instructions. Membranes were then stripped, reprobed with anti-β-Actin antibodies and exposed again to detect the internal β-Actin standard.

**Figure 4 molecules-25-04693-f004:**
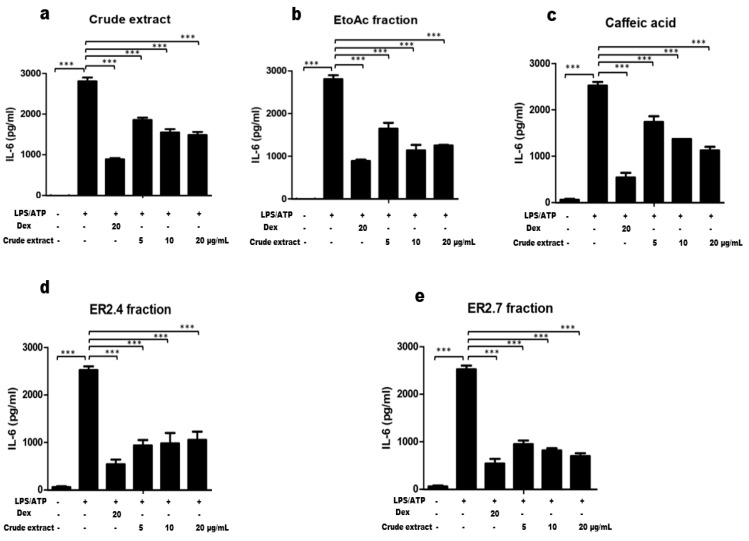
Inhibition of IL-6 production by *H. noldeae* constituents. Cells were pre-treated with crude extract (**a**), EtoAc fraction (**b**) and partially purified ER 2.4 (**d**) and ER 2.7 (**e**) fractions, and caffeic acid (**c**) for 3 h and stimulated by LPS (1 µg/mL) for 18 h. Dexamethasone (Dex) at 20 µg/mL was used as a control. Cell-free supernatants were harvested and stored at −70 °C until analysis. The concentration of IL-6 in the culture supernatant was measured using IL-6 ELISA kit (BD Biosciences) according to the manufacturer’s instructions. For each experiment, the amount of IL-6 produced was considered as 100% and the remaining treatments was calculated as (Treatment/LPS) × 100. The results are presented as the mean ± SD from a triplicate of the representative experiment. A one-way ANOVA, followed by Benforroni’s multiple comparison test, was used to compare the sample groups. *** stands for a *p* < 0.0001.

**Table 1 molecules-25-04693-t001:** Mobile phase gradient for the analytical High-Performance Liquid Chromatography (HPLC).

Time (min)	Mobile Phase Proportion (%)	
	Solvent A	Solvent B
0.0	0.0	100
1.0	3.0	97.0
45.0	40.0	60.0
55.0	40.0	60.0
56.0	60.0	40.0
66.0	60.0	40.0
67.0	0.0	100
82.0	0.0	100

**Table 2 molecules-25-04693-t002:** Gradient system used to isolate compounds on preparative HPLC.

Time (min)	Mobile Phase Proportion (%)	
	Solvent A	Solvent B
0.0	0.0	100
2.0	3.0	97.0
35.0	15.5	84.5
55.0	19.0	81.0
76.0	27.5	72.5
98.0	31.5	68.5
133.0	60.0	40.0
144.0	60.0	40.0

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
