# Peer review of "The Inhibition of NLRP3 Inflammasome and IL-6 Production by Hibiscus noldeae Baker f. Derived Constituents Provides a Link to Its Anti-Inflammatory Therapeutic Potentials"

_molecules, 2020, doi:10.3390/molecules25204693_

Round 1

Reviewer 1 Report

Traditional medicine for asthma in Rwanda use H. noldeae as a major constituent of their formula. In this study, the authors hypothesized that the extracts from this plant may hold the therapeutic features attributed to the traditional remedies. In order to explore this hypothesis, they isolated fractions and pure compounds which were tested for their efficacy in NLRP3 inflammasome activation, one of the major biological pathways linked to the development and severity of asthma. Their findings, partly support this notion, but need to be strengthened in order for the manuscript to be accepted for publication.

There are minor grammatical errors occurring throughout the manuscript and particularly in the introduction section.

There has been a peculiar selection of references, especially considering the extremely large collection of publications from renowned researchers in the field of asthma. Reference #2 is a non-english publication, something that is not accustomed to be used in international journals. Also, please update the asthma metrics to 2020, as reference #1 goes back to 2013.

Lines 70-77: The activation of NLRP3 is described in the introduction section in a rather simplified manner. The 1st and 2nd signals cannot only be PAMPs and DAMPs, as the authors state, respectively. Some of the references used provide a significantly more detailed description of this complex mechanism, so please revise accordingly.

Line 95: Change H. acetossela to H. acetosella.

Regarding figures 2-4, since LPS/ATP are added in compound-treated cells I recommend to designate it in the figure itself and not only in the legend.

e.g.:

LPS/ATP        -           +          +          +          +

Compound      -           -           5          10        20

Additionally, in regards to the statistics, it would be more accurate to mention the correspondence of stars to p values, rather than making a subjective statement like “ *** means very significant”.

No dose-dependent inhibition is observed in figure 2 for caffeic acid, ER2.4 and ER2.7 despite the fact that the authors are stating the opposite. The data and statistical interpretation leave no room for this claim.

Lines 176, 194 and 393: As far as I know, Invivogen is not part of Thermo Fisher and it doesn’t offer IL-1β ELISA kits. Do the authors mean Invitrogen, instead?

In figure 3, untreated cells are not included in the graph, so although the authors mention that LPS/ATP increased IL-1b secretion significantly (line 179), the reader cannot verify it.

More importantly, cytokine levels cannot be assessed adequately well, if absolute levels are not depicted. Please revise both figure 3 and 4 and show protein levels measured as per manufacturer’s recommendations and present them as pg/ml.

Lines 182-186: the statistics of figure 3 data fail to support the conclusions that the authors have reached regarding the different degree of effect observed between compounds and concentrations. Setting the LPS/ATP sample levels at 100% without any deviation, generates biased analyses between pretreated and non-pretreated samples. Revised figures which will depict absolute cytokine levels would probably increase the credibility of the results and conclusions. The same applies for figure 4. Except, for a trend shown by ER2.7, there are no dose-dependent effects observed for any compound.

Line 221: replace IL-1β with IL-6.

Line 258: Figure S.3 doesn’t provide the information stated.

I do not believe that Figure 5 is accurate.

The claim that ER2.7 and dexamethasone both target NF-kB holds no ground. As shown in figure 2, ER2.7 also blocks caspase-1 activation in a similar fashion to caffeic acid and ER2.4.

Reference #34 presents solid data that caffeic acid phenethyl ester blocks NLRP3-ASC interaction. But according to reference #33, blueberry extracts, including caffeic acid and quercetin, affect both the expression of murine Il6 and Il1b genes as well as the protein levels of the mature caspase-1.

Further experiments are required to support the authors’ hypotheses regarding the specificity of the compounds’ inhibitory actions. The simplest ones that could strengthen this study are proposed below:

(1) Pro-IL-1β protein levels could be measured intracellularly via ELISA or WB.

(2) IL-6 and IL-1β gene expression patterns could be quantified via qPCR.

(3) The formation of ASC specks could be observed via immunofluorescence.

(the use of a single compound concentration would be enough)

The proposed experiments can answer whether the compounds tested affect the transcriptional activity of NF-kB, the assembly of the NLRP3 inflammasome or the caspase-1 activity, which lies downstream.

General comment: according to the study’s results, crude extracts, fractions and pure compounds all show, more or less, the same degree of inhibition of caspase-1 activity. Why would anyone proceed with the labor-intensive isolation and purification of the compounds if the crude extracts offer almost the same effect?

Reviewer 2 Report

Given the previously reported anti-inflammatory properties of molecules from the Hebiscus genus in this work authors identify and extract several compounds from the relatively understudied H. noldeae and test their anti-inflamamtory potential by looking at the impact of these compunds and the crude extract itself on IL-1beta and IL-6 secretion from both mouse and human macrophage/monocyte cell line.

It is certainly an interesting observation that some of the compounds from H. noldeae (mainly ER2.4) appear to inhibit IL-1beta and IL-6 secretion. However, the data is quite limited for the conclusions drawn by the authors. Data is limited to looking at secretion of the two cytokines and caspase-1 ‘activity’ by Caspase-Glo Assay of supernatants (Promega).

I would recommend further experiments to clarify a ‘mechanism’ as inflammasome inhibition is not definitively demonstrated.

Major comments:

Have authors checked by Western or Real-Time PCR to check if compounds impact on the induction of Pro-IL-1 and Pro-caspase-1 transcript or NLRP3? Authors should do these experiments for IL-1beta and Caspase-1 in the very least. In addition, while viability assays were performed on cells treated with compound alone there is no data provided for inflammasome associated cell death, ‘pyroptosis’. This can be easily performed on supernatants using LDH assay (Promega).

The impact of these compounds is more than likely transcriptional given the significant and ‘early’ impact on IL-6. Authors could perform Westerns for p65 and/or IKKb as indicators of inhibition of NF-kB pathway. Has NFkB inhibition been shown for other plants in the genus?

Importantly other studies show that Caffeic acid inhibits the NLRP3 inflamamsome by direct interaction with ASC.  https://www.nature.com/articles/srep38622

Can authors confirm if the viability assays on the THP-1 with the different compounds were performed on PMA-differentiated THP-1s as for the rest of the paper??

Certain results not described as per data in figures. Please amend. Legend for figure 2 states:

Line 160 – 164: Figure 2. Authors state that ‘ dose dependent 161 inhibition of the caspase-1 activation could be observed with the purified Caffeic acid, ER2.4 and 162 ER2.7 fractions (Figure 2).’ However it is very clear from the data that a ‘possible’ dose dependency is only observed for ER2.4.

I do not agree that the work sheds light on a mechanism of action of H. noldeae however from what I can see in the databases it does provide the very first scientific based evidence for its anti-inflammatory effects and widespread use by traditional healers in Rwanda for a variety of ailments. (and that of some of its constituents) 

Minor comments:

Typo line 61: inflammatory, some letters in bold.

Line 73: please use correct symbol for kappa in NF-kappaB

Lin 71 – 75: Signal 1 via TLR4 also upregaultes Pro-Caspase-1 expression please include along with NLRP3 and pro-IL-1beta.

Lijne 76: A nice way of describing caspase-1 maturation is, ‘proximity induced autocleavage’ i.e. when so may pro-capase-1 molecules are in close contact with each other this induces cleavage and ‘maturation’.

Line 84 – 86: sentence referencing the use of Dexamethasone in SARS COVID-19 patients seems to be misplaced and does not flow in the context of the previous sentence about asthma. Please amend flow of paragraph.

Introduction: I feel like the content of the introduction needs to be flipped. The reader has no idea what context or why you are looking at inflammasomes until near the end of the introduction. It would read much better describing the problem of Asthma and treatments in LMICs first and then its makes sense why you are looking at inflammasomes and detailing all of the mechanisms of inflammasome.  

Typo line 130: identified, some letters in bold.

Author Response

Response to reviewer 2 Comments

Comments and Suggestions for Authors: Given the previously reported anti-inflammatory properties of molecules from the Hebiscus genus in this work authors identify and extract several compounds from the relatively understudied H. noldeae and test their anti-inflamamtory potential by looking at the impact of these compunds and the crude extract itself on IL-1beta and IL-6 secretion from both mouse and human macrophage/monocyte cell line.

It is certainly an interesting observation that some of the compounds from H. noldeae (mainly ER2.4) appear to inhibit IL-1beta and IL-6 secretion. However, the data is quite limited for the conclusions drawn by the authors. Data is limited to looking at secretion of the two cytokines and caspase-1 ‘activity’ by Caspase-Glo Assay of supernatants (Promega).

I would recommend further experiments to clarify a ‘mechanism’ as inflammasome inhibition is not definitively demonstrated.

Authors’ response: We thank the reviewer for this favorable view of our paper and agree to strengthen our findings in order to further support our conclusions. In this context, we have conducted two sets of experiments requested and which addressed (1) the impact of H. noldeae constituents on pyroptosis, and (2) the effects these compounds and plant fractions of interest on the expression of key determinants of the canonical NLRP3 inflammasome activation, such as the Pro-Caspase-1 and the Pro- IL-1β proteins. As results, we found that our tested samples revealed trends on pyroptosis (Figure S.6) similar to what was observed on IL-1β production. Secondly, western blot analysis revealed that Pro-IL-1β expression was not negatively affected by H. noldeae constituents, suggesting that the tested samples likely did not act on the priming step of the inflammasome activation. Consistently, we observed for example that cell pretreatment with ER2.4 and ER2.7 fractions revealed bold signals corresponding to the amount of Pro-IL-1β proteins (Figure 3B), suggesting that their cleavage was strongly inhibited. All these new findings support the notion that Hibiscus noldeae derived constituents act on inflammasome activation and likely do not impact the priming step but rather act on the second signal, namely the NLRP3 inflammasome activation, to block the cleavage of Pro-IL-1β. The manuscript has been revised accordingly.  

Major comments:

Comment 1: Have authors checked by Western or Real-Time PCR to check if compounds impact on the induction of Pro-IL-1 and Pro-caspase-1 transcript or NLRP3? Authors should do these experiments for IL-1beta and Caspase-1 in the very least. In addition, while viability assays were performed on cells treated with compound alone there is no data provided for inflammasome associated cell death, ‘pyroptosis’. This can be easily performed on supernatants using LDH assay (Promega).

Authors’ response: We thank the reviewer for these valuable comments. As recommended, we conducted both the Pyroptosis assay and the western blot (WB) experiments targeting both Pro-Caspase-1 and Pro-IL-1β proteins. Thanks to the western blot results on both Pro-Caspase-1 and Pro-IL-1β, the effects of our compounds and fractions of interest should be located at the level of the 2 axes of the NLRP3 inflammasome activation. For example, upon pretreatment of cells with ER2.4 and ER2.7 partially purified compound, the Pro-IL-1β production appeared induced as marked by the width of the WB-signals (Figure 3B). These observations suggest that both ER2.4 and ER2.7 fractions likely do not impact the priming step but rather act on the second signal, namely the NLRP3 inflammasome activation, to block the cleavage of Pro-IL-1β. However, the full mechanism of action will be determined after purification of these two fractions. Concerning the pyroptosis assays, we investigated the effect of our compounds and fractions of interest on inflammasome activation-induced cell death by measuring the release of lactate dehydrogenase (LDH) in the culture supernatants. As expected, the pretreatment of cells with the plant constituents of interest significantly inhibited ATP-induced pyroptosis (Figure S.6).

Comment 2: The impact of these compounds is more than likely transcriptional given the significant and ‘early’ impact on IL-6. Authors could perform Westerns for p65 and/or IKKb as indicators of inhibition of NF-kB pathway. Has NFkB inhibition been shown for other plants in the genus? Importantly other studies show that Caffeic acid inhibits the NLRP3 inflamamsome by direct interaction with ASC.  https://www.nature.com/articles/srep38622

Authors’ response: The western blot results (Figure 3B) suggested that ER2.4 and ER2.7 fractions did not impact the first inflammasome priming signal. Given the time constraint and limitation because of the COVID-19, the proposed westerns for p65 and/or IKKb as indicators of inhibition of NF-kB pathway will be performed in the future work aiming at determining the molecular mechanism of action of the purified molecules from the ER2.4 and ER2.7 fractions.

Comment 3: Can authors confirm if the viability assays on the THP-1 with the different compounds were performed on PMA-differentiated THP-1s as for the rest of the paper??

Authors’ response: THP-1-derived macrophages are less sensitive than the parent THP-1 monocytes. We surely investigated the cytotoxicity of our compounds of interest on them, but results are not shown in this study.

Comment 4: Certain results not described as per data in figures. Please amend. Legend for figure 2 states:

Line 160 – 164: Figure 2. Authors state that ‘dose dependent 161 inhibition of the caspase-1 activation could be observed with the purified Caffeic acid, ER2.4 and 162 ER2.7 fractions (Figure 2).’ However it is very clear from the data that a ‘possible’ dose dependency is only observed for ER2.4.

Authors’ response: The reviewer is fully right and the paragraph was amended as follows “The tested compounds revealed their maximal inhibitory activities at 5 µg/ml, except for the ER2.4 fraction that showed a dose-dependent inhibition of the caspase-1 activation” (see lines 161-167).

Comment 5: I do not agree that the work sheds light on a mechanism of action of H. noldeae however from what I can see in the databases it does provide the very first scientific based evidence for its anti-inflammatory effects and widespread use by traditional healers in Rwanda for a variety of ailments. (and that of some of its constituents).

Authors’ response: We agree with the reviewer and the sentence was amended as follows “Overall, our work does provide the very first scientific based evidence for Hibiscus noldeae anti-inflammatory effects and widespread use by traditional healers in Rwanda for a variety of ailments” (lines 35-37).

Minor comments:

Comment 6: Typo line 61: inflammatory, some letters in bold:

Authors’ response: The typing error has been corrected.

Comment 7: Line 73: please use correct symbol for kappa in NF-kappaB

Authors’ response: Corrected all over the text (see lines 74, 183, 229, 231).

Comment 8: Lin 71 – 75: Signal 1 via TLR4 also upregulates Pro-Caspase-1 expression please include along with NLRP3 and pro-IL-1beta.

Authors’ response: Thank you for the suggestion. The paragraph was edited accordingly.

Comment 9: Line 76: A nice way of describing caspase-1 maturation is, ‘proximity induced autocleavage’ i.e. when so many pro-capase-1 molecules are in close contact with each other this induces cleavage and ‘maturation’.

Authors’ response: Thank you for the comment. It has been corrected as recommended. Now written as follows “The second signal is an activation step that induces the assembly of the NLRP3/ASC/pro-caspase-1 complex, ending up in the proximity auto-cleavage and maturation of caspase-1 which, in turn, induces a caspase-1-mediated maturation of pro-IL-1β and release of active IL-1β ». (see lines 75-78).

Comment 10: Line 84 – 86: sentence referencing the use of Dexamethasone in SARS COVID-19 patients seems to be misplaced and does not flow in the context of the previous sentence about asthma. Please amend flow of paragraph.

Authors’ response: The following sentence was added to ease the flow of paragraph “Despite this resistance, the use of corticosteroids remains the basic treatment for respiratory diseases involving cytokine storms. A recent clinical trial report revealed that treatment with Dexamethasone significantly alleviated COVID-19 disease severity of hospitalized patients who were under invasive mechanical ventilation or oxygen. See lines 85-90.

Comment 11: Introduction: I feel like the content of the introduction needs to be flipped. The reader has no idea what context or why you are looking at inflammasomes until near the end of the introduction. It would read much better describing the problem of Asthma and treatments in LMICs first and then its makes sense why you are looking at inflammasomes and detailing all of the mechanisms of inflammasome.  

Authors’ response: We think that beyond understanding the scientific based evidence of using Hibiscus noldeae as medicinal plant in LMICs our focus on inflammasome activation as a key process in disease with inflammatory background could justify the structure of our introduction. For this reason, we prefer not to flip as suggested.

Comment 12: Typo line 130: identified, some letters in bold.

Authors’ response: The typing error has been corrected, see line 133.

Round 2

Reviewer 1 Report

I believe that the manuscript has been adequately improved and with the inclusion of new data, key questions have been addressed regarding the effect of H. noldeae fractions on the regulation of NLRP3 inflammasome activation. Therefore, I feel that the revised manuscript should be accepted for publication. 

Author Response

We thank the Reviewer for his positive feedback.

Reviewer 2 Report

I commend the authors on their efforts to perform additional experiments to address reviewer comments. Many of this reviewers concerns have been addressed. However, there is an outstanding issue regarding Figure 3 and an actual mechanism of action of fractions from H. noldeae. Pro-Caspase-1 is clearly inhibited on the protein level in western blots. The authors ignore this and simply refer to unaltered levels of Pro-IL-1beta. Capsase-1 is a critical component of the inflammasome and the reduced expression of this enzyme by fractions from H. noldeae indicates that ER2.4 and ER2.7 may in fact be acting to inhibit signal 1 of inflammasome activation by specifically blocking upregulation of Caspase-1. I appreciate further work and studies need to be performed to examine this fully but the authors must reflect this in their conclusion and discussion of their results. This reviewer is not satisfied with the current conclusions in the text. 

Author Response

We fully agree with the reviewer and have amended our conclusions.

The results descrition is corrected as follows in lines 214 to 222:

" We observed as shown in Figure 3B that cells pretreatment with ER2.4 and ER2.7 fractions led to the reduced Pro-Caspase-1 amount marked by the faint WB-signals. This suggest that our fractions of interest inhibited the expression of this protein by impacting the first priming step of the NLRP3 inflammasome activation or induced its degradation by a mechanism to be clarified. Moreover, the Pro-IL-1β production appeared induced as marked by the width of the WB-signals (Figure 3B). Knowing the critical role of Caspase-1 in the inflammasome activation, these observations suggest that the reduced expression of the Pro-Caspase-1 by both ER2.4 and ER2.7 fractions may in fact be acting to inhibit the upregulation of Caspase-1, hence block the Pro-IL-1β cleavage."

The discussion section is corrected as follows in lines 316 to 323:

"NLRP3 inflammasome-associated IL-1β production is a two-step process. Our findings on both pyroptosis assays (Figure S.6) and IL-1β production (Figure 3A) suggest that H. noldeae derived products likely act on the second signal, namely NLRP3 inflammasome activation, to block cell death (pyroptosis) and IL-1β production. However, the western blot results showed that ER2.4 and ER2.7 fractions repressed the expression of Pro-Caspase-1 (Figure 3B). Capsase-1 is a critical component of the inflammasome and the reduced expression of this enzyme by fractions from H. noldeae indicates that ER2.4 and ER2.7 may in fact be acting to inhibit signal 1 (priming step) of inflammasome activation by specifically blocking upregulation of Caspase-1. In our future research the mode of action of the molecules contained in these fractions will be elucidated."